# miR-185-5p Regulates Inflammation and Phagocytosis through CDC42/JNK Pathway in Macrophages

**DOI:** 10.3390/genes13030468

**Published:** 2022-03-07

**Authors:** Xirui Ma, Huifang Liu, Jing Zhu, Caoxu Zhang, Yajie Peng, Ziming Mao, Yu Jing, Fengling Chen

**Affiliations:** Department of Endocrinology and Metabolism, Shanghai Ninth People’s Hospital, Shanghai JiaoTong University School of Medicine, Shanghai 200025, China; raemxx@sjtu.edu.cn (X.M.); lhf_404@163.com (H.L.); zj761045@126.com (J.Z.); 17721295725@163.com (C.Z.); pengyj1997@126.com (Y.P.); maoziming@sjtu.edu.cn (Z.M.); jingyu_1031@126.com (Y.J.)

**Keywords:** microRNA, macrophage, inflammation, JNK, CDC42

## Abstract

Macrophage activation is an essential component of systemic chronic inflammation and chronic inflammatory diseases. Emerging evidence implicates miR-185-5p in chronic inflammation diseases. However, the regulatory role of miR-185-5p in macrophage pro-inflammatory activation has not been studied previously. Here, we identified that miR-185-5p was one of the top genes and effectively downregulated in two macrophage miRNA expression datasets from GEO. Under LPS stress, miR-185-5p overexpression reduced pro-inflammatory cytokine expression, suppressed phagocytosis in RAW264.7 macrophage. miR-185-5p inhibitors augmented pro-inflammatory effects of LPS in macrophage. Mechanically, miR-185-5p sponged and negatively regulated the protein expression of CDC42. Ablation of CDC42 with selective CDC42 inhibitor CASIN reversed the pro-inflammatory effect of miR-185-5p inhibitors through inhibiting MAPK/JNK pathways. Collectively, these data demonstrate that miR-185-5p exhibited anti-inflammatory functions in LPS-induced RAW264.7 macrophages at least partially through CDC42/JNK pathways. Our findings yield insights into the understanding of miR-185-5p-regulated network in macrophages inflammation, which is beneficial for exploring miRNA-protein interaction in atherosclerotic inflammation.

## 1. Introduction

Atherosclerosis is a chronic inflammation disease, which is characterized by the development of atherosclerotic plaques in the vascular wall [1]. Atherosclerosis-related cardiovascular disease has become a leading cause of death worldwide [2]. Pro-inflammatory macrophages largely reside in intima and plaque shoulder of both mouse and human atherosclerotic plaques based on single-cell RNA-Seq findings [3]. Macrophage infiltration leads to non-resolving inflammatory condition [4]. In advanced atherosclerotic lesions, defective phagocytosis of apoptotic macrophages and incomplete clearance of cell debris amplify inflammation, which increases the risk of plaque rupture [5]. Monocytes and macrophages are hyper-inflammatory before recruitment to plaques in patients with atherosclerosis. In a human monocyte metabolism study, monocytes isolated from individuals with atherosclerosis produce higher cytokine than do monocytes from healthy individuals after stimulation with lipopolysaccharide (LPS) and interferon-γ (IFN-γ) [6,7]. It indicates that the modulation of aberrant macrophage inflammatory responses is critical target in atherosclerosis.

LPS-Toll-like receptor 4 (TLR4)-mediated inflammatory response induces pro-inflammatory gene expression [8]. Enhanced production of IL6 and IL-1β by macrophages are key to augment inflammatory signaling, contributing to atherosclerosis acceleration [6,7]. Mitogen-activated protein kinase (MAPK) family members, including ERKs (extracellular-signal-regulated kinases), JNKs (jun amino-terminal kinases), and p38/SAPKs (stress-activated protein kinases), are one of the critical cell signal transduction cascades in inflammation response [8,9]. Myeloid cell-specific silencing of JNK reduces the expression of several pro-inflammatory macrophage marker genes in bone marrow–derived macrophages (BMDMs) after LPS treatment [10]. Silencing of JNK could dramatically inhibit macrophage phagocytosis [11,12]. Targeting JNK pathway would be a potential therapeutic approach in inflammation-related diseases [13].

Cell division cycle 42 (CDC42), a member of Rho small GTPase family, is one of the upstream regulators in MAPK pathways [14,15,16], which is involved in signaling to modulate JNK [9,17]. CDC42 has been proved to play a pro-migratory effect on endothelial cells and vascular smooth muscle cells during atherogenesis [18,19]. Recent study indicates that CDC42 could be engaged in pro-inflammatory responses in ageing. CDC42 activity-specific inhibitor (CASIN) injection extends lifespan and decreases circulating inflammatory cytokines, such as IL-1β in aged female C57BL/6 mice [20]. However, whether CDC42 plays a functional role in macrophage inflammation, and what exactly the role may be, remains to be elucidated.

MicroRNAs (miRNAs) are small endogenous noncoding RNAs that mediate gene expression through targeting transcriptional, posttranscriptional, or translational processes [21]. Many studies implicate miR-185-5p in chronic inflammation diseases, such as non-alcoholic fatty liver disease (NAFLD), diabetes, and atherosclerosis [22,23,24,25]. Lower expression of miR-185-5p is observed in the plasma of diabetic patients. miR-185-5p inhibits β-cell dysfunction through targeting SOCS3 [24]. In high-fat diet C57BL/6J mice, miR-185-5p could help to reduce liver steatosis and ameliorate insulin resistance [23], which indicates a protective role of miR-185-5p in metabolic diseases. In atherosclerosis, Wang and Jiang demonstrate miR-185-5p is pro-atherogenic through modulating hepatic cholesterol metabolism in ApoE−/− mice [22,25].

To study the effect of miR-185-5p in atherosclerotic inflammation, we take advantage of two GEO datasets, GSE87718 (miRNA expression profiling of BMDMs from macrophage-specific Dicer knockout vs. Dicer wildtype ApoE−/− mice) and GSE143845 (miRNA profiling in RAW264.7 cells treated with or without LPS and IFN-γ). miR-185-5p is observed as one of the top differentially expressed miRNAs (DEMs) correlated with Dicer-regulated atherosclerosis development and macrophage inflammation. Here, we use LPS-induced RAW264.7 macrophage model to activate TLR4-mediated inflammation. Our results show that miR-185-5p inhibits inflammation and phagocytosis at least partially through targeting CDC42/JNK pathways.

## 2. Materials and Methods

### 2.1. miRNA Microarrays and Screening of DEMs

Key words (“macrophage” and “miRNA”) were used to screen miRNA datasets in the Gene Expression Omnibus (GEO) database (https://www.ncbi.nlm.nih.gov/gds, last accessed on 29 October 2021). GSE87718 [26] and GSE143845 [27] were selected for bioinformatic analysis. GEO2R (http://www.ncbi.nlm.nih.gov/geo/geo2r/, last accessed on 29 October 2021) is an interactive web tool. We applied GEO2R to screen DEMs in GSE87718 and GSE143845. Here, a *p* value < 0.05 and |fold change (FC)| > 1 were set as a threshold to identify DEMs in GSE87718, adjusted *p* value < 0.01 and |Log FC| > 2 were set as a threshold to identify DEMs in GSE143845. Heatmaps were made by using RStudio. Volcano plots were analyzed by online tool Hiplot (https://hiplot.com.cn/, last accessed on 31 October 2021). Detailed information about the two datasets is listed in Table 1.

### 2.2. Cell Culture and Treatment

Murine macrophage RAW264.7 was brought from FuHeng BioLogy (Shanghai, China). RAW264.7 cells were cultured in Dulbecco’s modified Eagle medium (Basal Media, Shanghai, China) with 10% fetal bovine serum (Sigma, St. Louis, MO, USA) at 37 °C with 95% air, 5% CO_2_. To build cell model of inflammation, cells were exposed to 100 ng/mL LPS (Sigma, St. Louis, MO, USA) for 24 h. Cells were pre-exposed to 1 μM CASIN (Sigma, St. Louis, MO, USA) for 6 h to selectively inhibit CDC42 activity, and pre-treated with 1 μM and 5 μM selective JNK inhibitor SP600125 (Sigma, St. Louis, MO, USA) for 6 h to selectively inhibit JNK pathway; DMSO (dimethyl sulfoxide) was treated as controls.

### 2.3. RNA Isolation and Real-Time Quantitative Polymerase Chain Reaction (RT-qPCR)

Total RNA of cells was extracted by using TRIzol Reagent (Invitrogen, Gaithersburg, MD, USA). The cDNA synthesis kit (Takara, Kyoto, Japan) was used for reverse transcription of mRNA. The miRNA 1st Strand cDNA Synthesis (by stem-loop) Kit (Vazyme, Nanjing, China) was used to reverse transcribe miRNA. RT-qPCR was performed using the ABI QuantStudio 12K Flex Real-time PCR System (Life Technologies, Gaithersburg, MD, USA). TB Green^®^ Premix Ex Taq™ (Takara, Kyoto, Japan) was used for mRNA RT-qPCR, and miRNA Universal SYBR qPCR Master Mix (Vazyme, Nanjing, China) was used for miRNA RT-qPCR. Relative gene expression was determined using the 2^-ΔΔCt^ cycle threshold method [28]. The mRNAs were normalized to endogenous control gene *Actb*, *U6* was used as the internal control for miRNAs. The sequences of primers used are described in Table 2.

### 2.4. Cell Transfection

The mmu-miR-185-5p mimics and mmu-miR-185-5p inhibitors were chemically synthesized by Ribobio Biotechnology (Guangzhou, China). RAW264.7 macrophages were transfected with 50 nM mmu-miR-185-5p mimics or 150 nM mmu-miR-185-5p inhibitors by using TransExcellentTM-siRNA (Cenji, Shanghai, China) according to the manufacturer’s protocol. Cells were transfected with equal amounts of non-targeting normal control (NC) miR NC or anti-miR NC. The sequences of miR-185-5p mimics, miR NC, miR-185-5p inhibitors, and anti-miR NC are listed in Table 2.

### 2.5. Prediction for Target Genes of mmu-miR-185-5p

ENCORI (https://starbase.sysu.edu.cn/index.php, last accessed on 16 October 2021), miRDB (http://mirdb.org/, last accessed on 17 October 2021), and miRWalk (http://mirwalk.umm.uni-heidelberg.de/, last accessed on 4 October 2021) are fine-designed online bioinformatic databases of predicting the RNA–protein interaction networks. The target genes of mmu-miR-185-5p were predicted and intersected by using three tools. Here, Target Score > 89 in miRDB, binding potential = 1 and energy < −20 in miRWalk were set as a threshold for identifying target mRNAs. Venn diagram was utilized to display the overlapping target genes.

### 2.6. Dual Luciferase Reporter Assays

To construct reporter plasmids, fragments of CDC42 3′ UTR containing predicted binding sites for mmu-miR-185-5p (wildtype, CDC42-WT) or fragments containing mutated binding sequences (mutated, CDC42-MUT) were inserted into pmirGLO dual luciferase miRNA target expression vector (Promega, Madison, WI, USA). CDC42-WT and CDC42-MUT pmirGLO plasmids were respectively co-transfected with miR-185-5p mimics or miR NC in 293T cells using Lipo2000 (Invitrogen, Gaithersburg, MD, USA). About 48 h later, luciferase activity was measured using dual luciferase reporter assay system (Promega, Madison, WI, USA), relative firefly luciferase activities were detected by a microplate reader (Bio-Tek, Winooski, VT, USA), which was normalized to renilla luciferase.

### 2.7. ELISA Assays

After stimulating cells with 100 ng/mL LPS for 24 h, cell culture supernatants were centrifuged at 300× *g* for 10 min to remove cell debris. ELISA kits were used for mouse IL6 (Multi Sciences, Hangzhou, China) and mouse IL-1β (Multi Sciences, Hangzhou, China) detections according to the manufacturer’s protocols. Optical density (OD) values at 450 nm and 570 nm were detected by a microplate reader (Bio-Tek, Winooski, VT, USA). Results were expressed as pg/mL.

### 2.8. Phagocytosis Assays

Phagocytosis was measured by Vybrant Phagocytosis Assay Kit (Thermo Fischer Scientific, Waltham, MA, USA) [29]. Cells were plated in black 96-well plates and incubated with FITC-labeled *E. coli* BioParticles for 2 h at 37 °C, then immediately incubated with trypan blue for 1 min at room temperature. Subsequently, cells were fixed with paraformaldehyde for 15 min and stained with DAPI. Cells were imaged using a fluorescence microscope Nikon Eclipse Ti-E (Nikon, Tokyo, Japan). The mean fluorescence intensity was analyzed by ImageJ software.

### 2.9. Western Blot

Total protein was extracted by 1× SDS protein lysis buffer supplemented with protease inhibitors (MCE, Monmouth Junction, NJ, USA) and phosphatase inhibitors (MCE, Monmouth Junction, NJ, USA). Total protein was separated in 12% SDS-PAGE and transferred to 0.45 μm PVDF membranes (EMD Millipore, Burlington, MA, USA). After blocking with 5% bovine serum albumin, the membranes were incubated respectively with Rabbit-anti Cdc42 (CST, Danvers, MA, USA), JNK (CST, Danvers, MA, USA), phospho-JNK (CST, Danvers, MA, USA), p44/42 MAPK (ERK1/2) (CST, Danvers, MA, USA), phospho-p44/42 MAPK (ERK1/2) (CST, Danvers, MA, USA), p38 (CST, Danvers, MA, USA), phospho-p38 (CST, Danvers, MA, USA), p65 (CST, Danvers, MA, USA), phospho-p65 (CST, Danvers, MA, USA), HSP90 (CST, Danvers, MA, USA), β-Tubulin (CST, Danvers, MA, USA) and GAPDH primary antibodies (CST, Danvers, MA, USA), then conjugated with Goat anti-Rabbit secondary antibody (Invitrogen, Gaithersburg, MD, USA). Proteins were visualized by Odyssey CLx Infra-Red Imaging system (Odyssey CLx, Lincoln, NE, USA).

### 2.10. Statistical Analyses

Student’s t test was used to analyze the statistical differences between two groups. Values were presented as means ± standard deviation (SD). Scatter histogram was performed using GraphPad Prism software (GraphPad Software Inc., La Jolla, CA, USA). Comparisons between multiple groups were evaluated by one-way ANOVA. Western blot quantification and fluorescence intensity quantification were analyzed and determined by ImageJ. *p* value < 0.05 was regarded as statistically significant.

## 3. Results

### 3.1. Identification of the Candidate DEM

Two expression profiles (GSE87718 and GSE143845) were selected from the GEO database [26,27]. With the threshold of *p* value < 0.05 and |fold change (FC)| > 1, 129 DEMs (3 upregulated and 126 downregulated) were identified in GSE87718 (Figure 1A,B). Total of 34 DEMs (14 upregulated and 20 downregulated) were identified in GSE143845 after screening with the threshold of an adjusted *p* value < 0.01 and |log FC| > 2 (Figure 1C,D). Detailed information of the GSE87718 and GSE143845 datasets is shown in Appendix A. The Venn diagram indicated seven overlapping DEMs in the two expression profile datasets (Figure 1E). Among these seven DEMs, miR-185-5p, named miR-185 previously (Appendix A), was the top overlapping DEMs and consistently downregulated in two datasets (Figure 1F). Expression values of each sample in two datasets were listed in Appendix A. To confirm whether miR-185-5p changed in response to TLR4 signaling in vitro, we only used LPS, instead of LPS&INF-γ to specifically trigger TLR4 signaling in RAW264.7 mouse macrophage cell line. As the result, miR-185-5p was significantly downregulated by LPS (Figure 1G).

### 3.2. miR-185-5p Overexpression Inhibits LPS-Induced Inflammation and Phagocytosis

To examine the function of miR-185-5p, we first transfected miR-185-5p mimics in RAW264.7 macrophages. The results indicated miR-185-5p was effectively upregulated relative to miR NC (Figure 2A). LPS-induced mRNA expression of *Il1b* and *Il6* was significantly suppressed in RAW264.7 macrophages with miR-185-5p mimics transfection (Figure 2B). The mRNA expression of *Tnf* was not altered (Figure 2B). In parallel, IL-1β and IL6 levels were markedly decreased by overexpressing miR-185-5p (Figure 2C).

TLR4 receptor complex stimulated by LPS regulates the expression of phagocytic receptors, which help to internalize large particles into phagosomes [10,30]. We explored whether miR-185-5p regulated fluorescein-labeled *Escherichia coli* (*E. coli*) bioparticles phagocytosis in RAW264.7 cells. As indicated in Figure 2D,E, upregulating miR-185-5p markedly suppressed LPS-induced macrophage phagocytosis compared with miR NC. These results indicate that miR-185-5p overexpression inhibits macrophage inflammation and phagocytosis.

### 3.3. miR-185-5p Suppression Promotes LPS-Induced Inflammation and Phagocytosis

Next, we examined if miR-185-5p knockdown could promote inflammation and phagocytosis. Figure 3A confirmed the transfection efficiency of miR-185-5p inhibitors in RAW264.7 macrophages when compared to anti-miR NC. As indicated, LPS-induced mRNA expression of *Il1b* and *Il6* was markedly increased after transfecting miR-185-5p inhibitors compared to anti-miR NC (Figure 3B). IL-1β and IL6 productions were improved consistently (Figure 3C). In addition, miR-185-5p knockdown effectively increased LPS-stimulated macrophage phagocytosis relative to anti-miR NC (Figure 3D,E). Thus, in RAW264.7 macrophages miR-185-5p inhibitors can further augment the pro-inflammatory effects of LPS.

### 3.4. miR-185-5p Regulates the Activation of MAPK Pathways

NF-κB and MAPK pathways play important role on LPS-activated inflammation in macrophages, we next examined the relation between miR-185-5p and the phosphorylation of key signaling proteins in NF-κB and MAPK pathways. Interestingly, miR-185-5p overexpression reduced the aberrant activation of phosphorylation in JNK, ERK1/2, and p38 (Figure 4A,B) compared to miR NC. In parallel, knocking down miR-185-5p led to increased activation of JNK, ERK1/2, and p38 signaling pathways after LPS treatment (Figure 4C,D). The LPS-induced phosphorylation of NF-κB p65 was not markedly changed by mediating miR-185-5p. These data suggest that miR-185-5p acts as a negative regulator in MAPK signaling pathway in LPS-activated macrophages.

### 3.5. CDC42 Is a Direct Target of miR-185-5p

Interaction with proteins is a major mechanism by which miRNAs regulate gene expression [31]. Therefore, to gain insights into the mechanism of miR-185-5p-regulated macrophage inflammation, downstream target genes of miR-185-5p were predicted by using ENCORI, miRWalk, and miRDB. We observed 16 overlapping genes (Figure 5A). Among these, we focused on those known to be involved in MAPK pathways, and finally selected *Cdc42* as a promising downstream candidate. To confirm the correlation between miR-185-5p and *Cdc42*, we transfected miR-185-5p mimics or miR-185-5p inhibitors and their normal controls in RAW264.7 macrophages. Although miR-185-5p failed to alter mRNA expression of *Cdc42* (Figure 5B), the protein level of CDC42 was negatively correlated with miR-185-5p expression (Figure 5C–F).

The predicted miR-185-5p target sites at CDC42 3′ UTR, shown in Figure 5G and Appendix A, were highly conserved across different species. We conducted dual luciferase report assays to verify the direct binding sites of CDC42 3′ UTR and miR-185-5p. In Figure 5H, miR-185-5p mimics greatly reduced wild type CDC42 3′ UTR luciferase activity, not mutants. These results indicate miR-185-5p can directly interact with CDC42 at protein level.

### 3.6. CASIN Modulates miR-185-5p-Mediated Pro-Inflammatory Phenotype Partially through Targeting MAPK JNK Pathways

Given the negative correlation between miR-185-5p and CDC42 described above, we hypothesized that miR-185-5p plays an anti-inflammatory role in macrophage by targeting CDC42. To test this, we used a selective small-molecule inhibitor of CDC42 named CASIN [32], to assess the effect of CDC42 inhibition on macrophage inflammation in RAW264.7 cells. CASIN rescued the mRNA level of *Il1b* and *Il6* (Figure 6A) in LPS-activated macrophages with miR-185-5p inhibitors transfection. Consistently, miR-185-5p inhibitors-mediated cytokine secretion of IL-1β and IL6 was also reversed by CASIN treatment (Figure 6B). Furthermore, CASIN also reduced phagocytosis which was previously increased in LPS-induced macrophages with miR-185-5p suppression (Figure 6C,D). Thus, miR-185-5p inhibitors promoted pro-inflammatory responses, and notably this effect was markedly reversed by inhibiting CDC42.

Next, we examined the CASIN’s effect on critical protein phosphorylation in MAPK signaling pathways. Under LPS, miR-185-5p inhibitors led to increased activities of JNK, ERK1/2, and p38 signaling pathways in RAW264.7 cells. Interestingly, CASIN significantly suppressed the phosphorylation of JNK, but enhanced the phosphorylation of ERK1/2 and p38 at the same time (Figure 6E,F). Following these results, we further examine whether miR-185-5p-CDC42 interaction regulated macrophage inflammation through JNK signaling. We observed that after LPS treatment, selective JNK inhibitor SP600125 exerted inhibitory effects on miR-185-5p inhibitors-enhanced cytokine secretion (Figure 6G). Thus, we believed miR-185-5p-CDC42 interaction can regulate, in part, the macrophage pro-inflammatory response though MAPK JNK pathway.

## 4. Discussion

Macrophages are key players in atherosclerotic inflammation [31]. Emerging evidence implicates that miRNAs in LPS induced macrophage inflammation. However, the regulatory role of miR-185-5p in macrophage pro-inflammation has not been studied previously. In this study, we showed a novel anti-inflammatory role of miR-185-5p in vitro. Mechanically, miR-185-5p suppresses LPS-induced inflammatory response by partially targeting CDC42/JNK, consequently alleviating macrophage inflammation (Figure 7).

We select miRNA profiling in Dicer knockout ApoE−/− mice, because Dicer is one of the key enzymes during miRNA biogenesis [33]. Macrophage-specific knockout of Dicer exhibits a pro-atherosclerotic phenotype with larger necrotic core in ApoE−/− mice. Pro-inflammatory gene expression including IL-1β and CCL2 is enhanced in lesional macrophages [26]. We observed that in absence of Dicer there was a huge decrease of miRNA expression in GSE87718, which is consistent with these findings in vitro Dicer knockout cells [34]. Young-Kook Kim and colleagues identify Dicer ablation can cause a depletion of 96% detected miRNA species [34]. miR-185-5p is significantly downregulated in the miRNA expression profiling of BMDMs in Dicer knockout compared to wildtype ApoE−/− mice.

Our results using mouse macrophages cell line confirmed an anti-inflammatory role for miR-185-5p. miR-185-5p mimics transfection decreased LPS-stimulated *Il1b* and *Il6* in RAW264.7 macrophages. Previous reports show similar evidence that in human umbilical vein endothelial cells (HUVECs) LPS effectively decreases the expression of miR-185-5p, and miR-185-5p overexpression exhibits anti-inflammatory effects via inhibiting Il6 and Ccl2 [35]. IL-1β is a powerful mediator of atherosclerosis development. Once activated, IL-1β induces the expression of another pro-inflammatory cytokine IL6 in macrophages, triggering the synthesis of C-reactive protein (CRP) [36]. Targeting IL-1β-IL6-CRP axis with Canakinumab significantly prevents atherosclerosis-related cardiovascular events in multi-center clinical trial [37]. Macrophage inflammatory response alleviated by miR-185-5p suppresses the key driver of atherosclerotic inflammation. Conflicting study reports a pro-atherosclerotic role of miR-185-5p in ApoE−/− mice by negatively modulating hepatic cholesterol metabolism [25]. However, evidence in Wang and Zhan’s study demonstrates miR-185-5p is protective toward cholesterol homeostasis in hepatocytes, consequently decreasing lipid synthesis in liver of C57BL/6 mice fed with high fat diet. Additionally, they indicate miR-185-5p increases insulin sensitivity by upregulating the insulin receptor substrate-2, which in turn activates the PI3K/AKT signaling [23].

Mechanically, miRNAs have been identified to interact with 3′ UTR of mRNAs thereby negatively regulating gene expression [33]. For example, miR-155 interacts with transcription factor Ets2 to modulate pro-inflammatory response in LPS-induced RAW264.7 [38]. We found miR-185-5p interacts with CDC42, which was consistent with previous findings [37]. CDC42 is critical for fetal survival that *Cdc42*-knockout mice are embryonic lethal [15,39]. CDC42 regulates diverse signaling pathways and cellular functions. The pro-migratory effect of CDC42 on endothelial cell and vascular smooth muscle cell has been demonstrated widely in atherogenesis [18,19]. Our novel results indicated that inhibition of CDC42 by sponging miR-185-5p effectively suppressed cytokine secretion and phagocytosis, which supported a pro-inflammatory role on CDC42. miR-185-5p might regulate CDC42 at post-translational level. Post-translational modifications are important for regulating protein function, localization, and stability. More investigations are needed to uncover the underlying mechanism. CDC42 is involved in signaling to the JNK modules [9,17]. In our study, ablation of CDC42 with CASIN decreased the secretion of IL-1β and IL6 and suppressed the phosphorylation of JNK, even though phosphorylated ERK1/2 and phosphorylated p38 were augmented in miR-185-5p knocking down macrophages. Earlier findings show that silencing CDC42 significantly enhances phosphorylation of p38 and ERK1/2 in human skin fibroblasts [40]. The effect of CDC42 on macrophage inflammation is rather more complicated than we have expected. Therefore, the regulatory effect of miR-185-5p/CDC42 on inflammation resolution is partially dependent on JNK pathways, not ERK1/2 and p38 pathways.

TLR4 activation enhances phagocytic receptors [10]. We for the first time shows a negative role of miR-185-5p/CDC42 on macrophage phagocytosis. CDC42 is quite important for phagocytosis [41,42]. In a subline of RAW 264.7 cells, intact RAC1 and CDC42 function serves as essential effectors of phagocytosis [42]. However, phosphoinositide 3-kinase enables phagocytosis of large particles in macrophages through inactivating Rac/Cdc42 [43]. In addition, the effect of phagocytosis is quite complicated in atherosclerosis. Enhanced phagocytosis improves lipid uptake and foam cell formation, enables lesion formation, which is protective in the early stage of atherosclerosis [31]. In advanced atherosclerosis, incomplete clearance and defective phagocytosis give rise to necrotic cores, which might contribute to plaque instability and plaque rupture [44]. Thus, further in vivo studies are needed to determine whether miR-185-5p plays diverse role in different stages of atherosclerosis.

Although we have proved that miR-185-5p can directly target CDC42, miRNA regulates cell functions through multitargets. There are also other miR-185-5p-regulated targets that existed in macrophage inflammatory activation. We found p21-activated protein kinases PAK4 may be another potential target for miR-185-5p with high prediction score in the predicting list of TargetScan. PAK4 has been proved to be a switch between caspase-8-modulated apoptosis and NF-κB in TNF-α-induced hepatocarcinoma cells [45]. We could not rule out other potential mechanisms involved in miR-185-5p-mediated macrophage inflammation. On the other hand, our study has some limitations. We demonstrated the effect of miR-185-5p in macrophage phagocytosis in vitro. Phagocytosis is an important function in macrophage and plays diverse roles in different stages of atherosclerosis [31]. In vivo study is urgently needed to illustrate whether miR-185-5p has an effect on macrophages infiltration and lesional inflammation in atherosclerosis. In addition, whether the regulatory effects of CDC42 on ERK1/2 and p38 are involved in atherosclerotic inflammation is not clear. In previous findings it was shown that, CDC42-ERK1/2 interaction is related to CCL2 secretion, which is a strong chemokine in macrophage recruitment [40]. The role of miR-185-5p on CCL2 expression needs further investigation.

## 5. Conclusions

In this study we showed that miR-185-5p exhibited anti-inflammatory functions at least in part through targeting CDC42/JNK pathways in LPS-induced macrophages. These findings yield insights into the understanding of miR-185-5p-regulated network in macrophages inflammation, which is beneficial for exploring the mechanism of macrophage inflammation in atherosclerosis.

## Figures and Tables

**Figure 1 genes-13-00468-f001:**
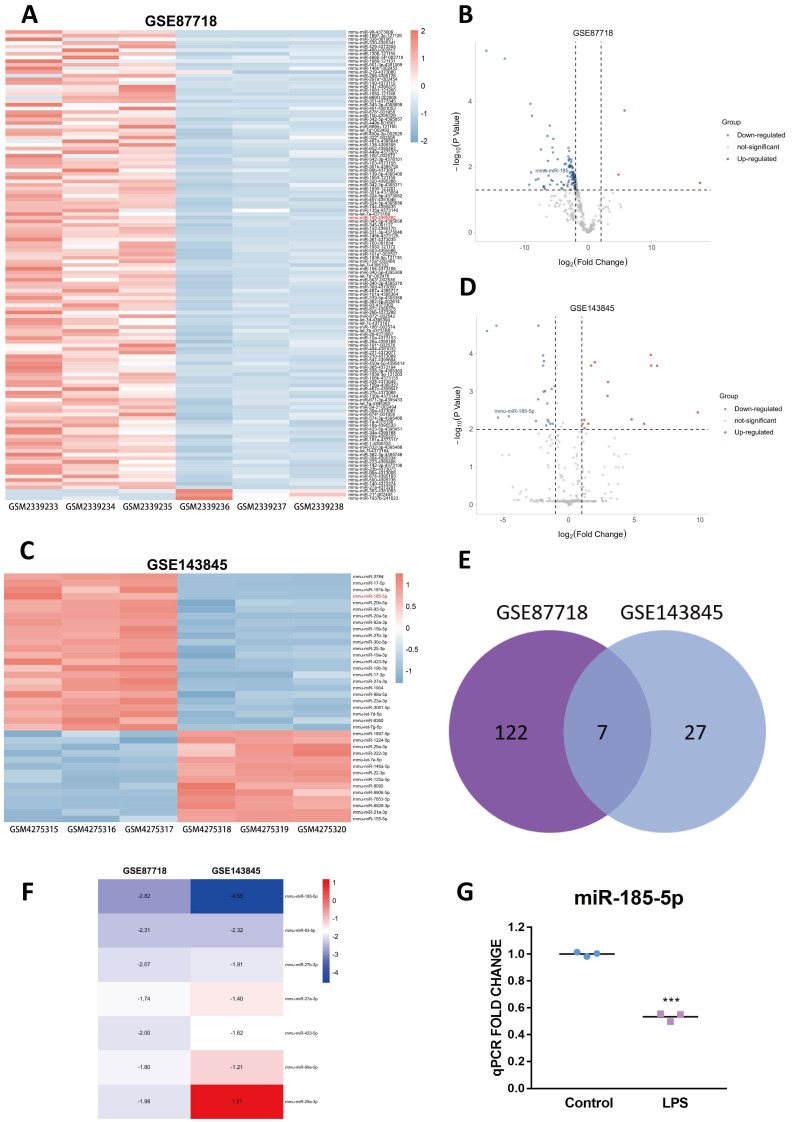
Identification of the candidate DEM. The heatmap (**A**) and volcano plot (**B**) of miRNAs in GSE87718, the screening threshold are *p* value < 0.05 and | FC| > 1. The heatmap (**C**) and volcano plot (**D**) of miRNAs in GSE143845, the screening threshold are adjusted to *p* value < 0.01 and |Log FC| > 2. The Venn diagram (**E**) and FC (GSE87718) and LogFC (GSE143845) heatmap (**F**) of overlapping DEMs. Abbreviations: DEMs, differentially expressed microRNAs. (**G**) The expression of miR-185-5p in RAW264.7 macrophages with or without 100 ng/mL LPS for 24 h. Mean ± SD; *n* = 3; *** *p* < 0.001.

**Figure 2 genes-13-00468-f002:**
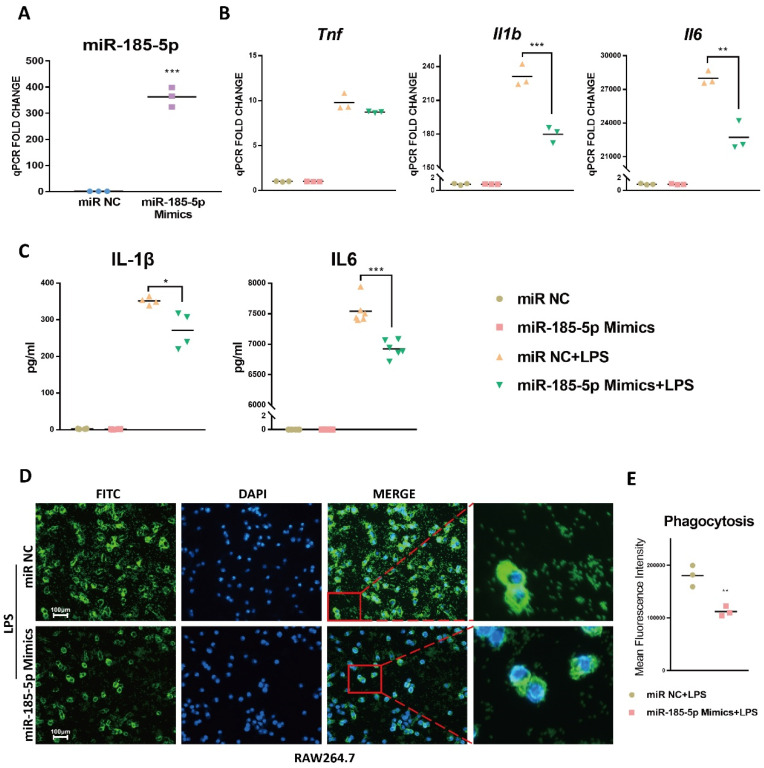
miR-185-5p overexpression inhibits inflammation and phagocytosis. (**A**) The transfection efficiency in RAW264.7 macrophages with 50 nM miR-185-5p mimics transfections vs. miR NC by RT-qPCR. (**B**) Gene expressions (RT-qPCR) and (**C**) cytokine secretions (ELISA) in LPS-induced RAW264.7 macrophages transfected with 50 nM miR-185-5p mimics or miR NC. (**D**) Representative images showing phagocytosis of fluorescein-labeled *E. coli* bioparticles (FITC-Green, DAPI-Blue) in RAW264.7 macrophages with 50 nM miR-185-5p mimics or miR NC transfection and 100 ng/mL LPS for 24 h. The scale bars correspond to 100 μm. (**E**) Quantification of mean fluorescence intensity of phagocytosis. Mean ± SD; *n* = 3–6; * *p* < 0.05; ** *p* < 0.01; *** *p* < 0.001.

**Figure 3 genes-13-00468-f003:**
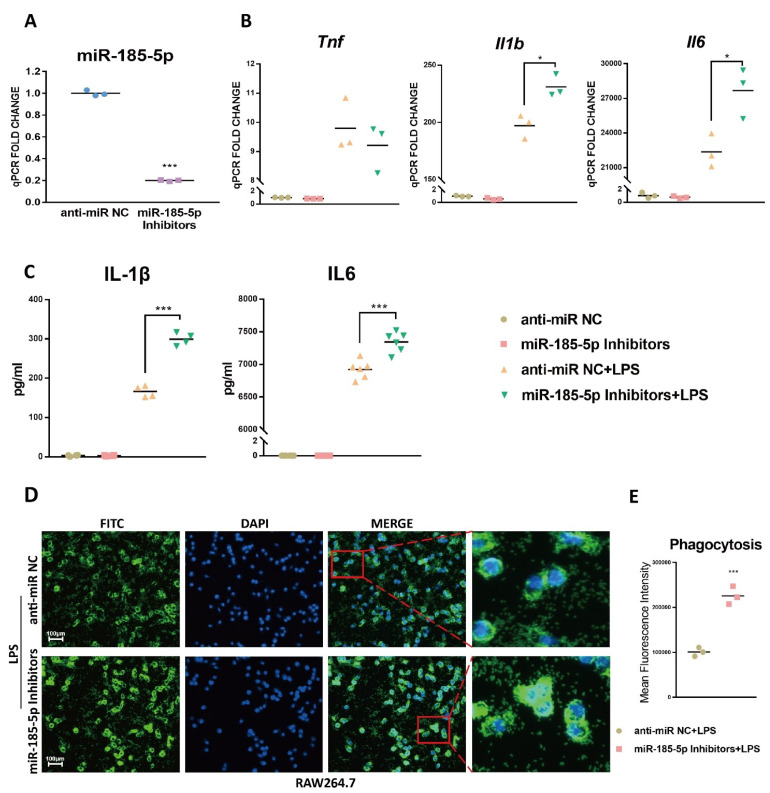
miR-185-5p suppression promotes inflammation and phagocytosis. (**A**) The transfection efficiency in RAW264.7 macrophages with 150 nM miR-185-5p inhibitors transfections vs. nontargeting control (anti-miR NC) by RT-qPCR. (**B**) Gene expressions (RT-qPCR) and (**C**) cytokine secretions (ELISA) in LPS-induced RAW264.7 macrophages transfected with 150 nM miR-185-5p inhibitors or anti-miR NC. (**D**) Representative images showing phagocytosis of fluorescein-labeled *E. coli* bioparticles (FITC-Green, DAPI-Blue) in LPS-induced RAW264.7 macrophages with 150 nM miR-185-5p inhibitors or anti-miR NC transfection. The scale bars correspond to 100 μm. (**E**) Quantification of mean fluorescence intensity of phagocytosis. Mean ± SD; *n* = 3–6; * *p* < 0.05; *** *p* < 0.001.

**Figure 4 genes-13-00468-f004:**
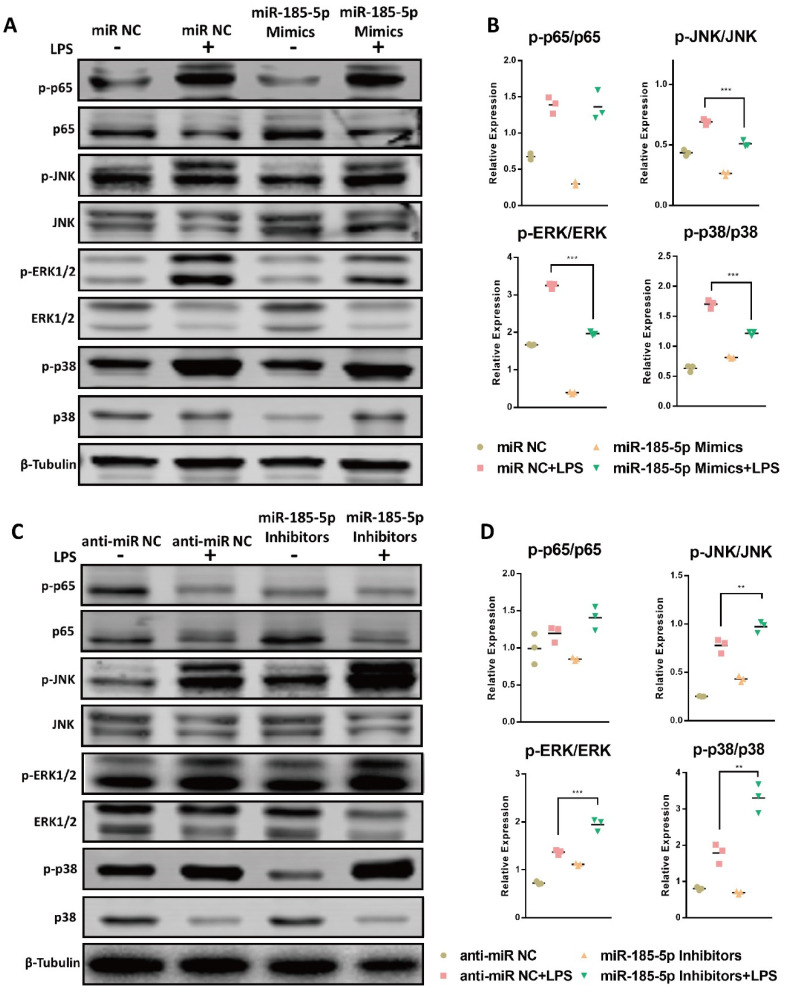
miR-185-5p regulates the activation of MAPK pathways. (**A**) Effects of miR-185-5p overexpression on the total and phosphorylation of p65, JNK, ERK1/2, and p38 in macrophages after treatment with or without 100 ng/mL LPS for 24 h, as determined by Western blot. (**B**) Quantification of phosphorylation and total proteins in Western blots in LPS-induced macrophages with miR-185-5p overexpression or miR NC. (**C**) Effects of miR-185-5p knockdown on the activation of the total and phosphorylation of p65, JNK, ERK1/2, and p38 in macrophages after treatment with or without 100 ng/mL LPS for 24 h, as determined by Western blot. (**D**) Quantification of phosphorylation and total proteins in Western blots in LPS-induced macrophages with miR-185-5p inhibition or anti-miR NC. Mean ± SD; *n* = 3; ** *p* < 0.01; *** *p* < 0.001.

**Figure 5 genes-13-00468-f005:**
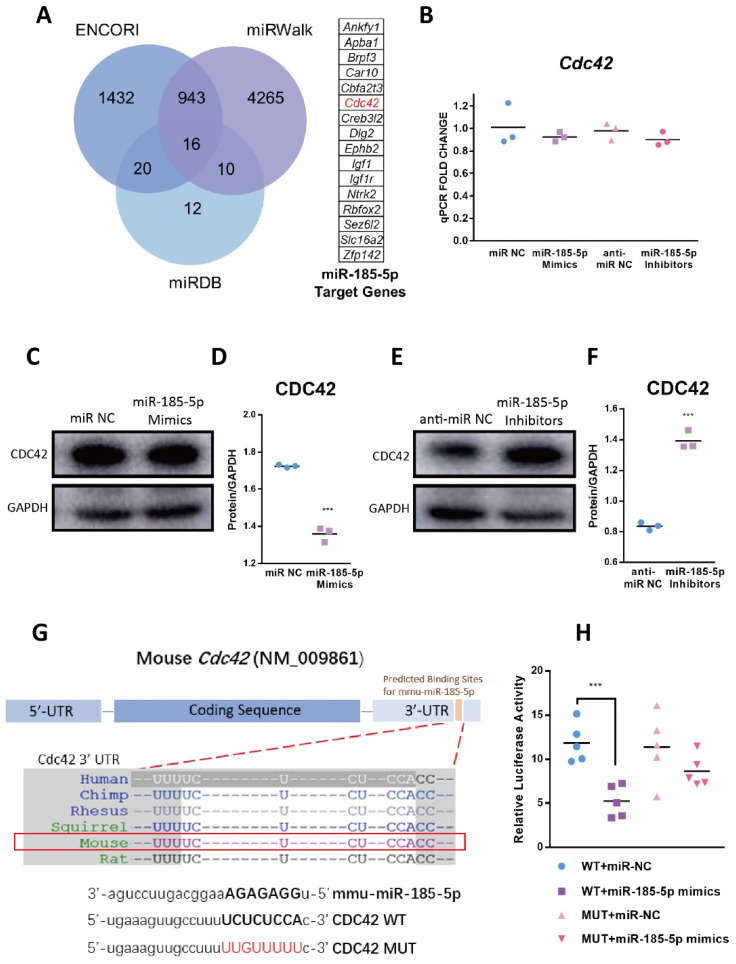
CDC42 is a direct target of miR-185-5p. (**A**) The intersection of miR-185-5p downstream target genes in ENCORI, miRWalk, and miRDB. Gene expression (**B**) and protein level (**C**,**E**) of CDC42 in RAW264.7 macrophages 48 h after transfection of miR-185-5p mimics or miR-185-5p inhibitors and miR NC or anti-miR NC. (**D**,**F**) Quantification of protein expression in Western blots. (**G**) Bioinformatic prediction of miR-185-5p target sites within the CDC42 3′ UTR. (**H**) Dual luciferase reporter assays examined miR-185-5p target sites within the CDC42 3′ UTR in 293T cells, the relative luciferase activity was measured. Mean ± SD; *n* = 3–5; *** *p* < 0.001.

**Figure 6 genes-13-00468-f006:**
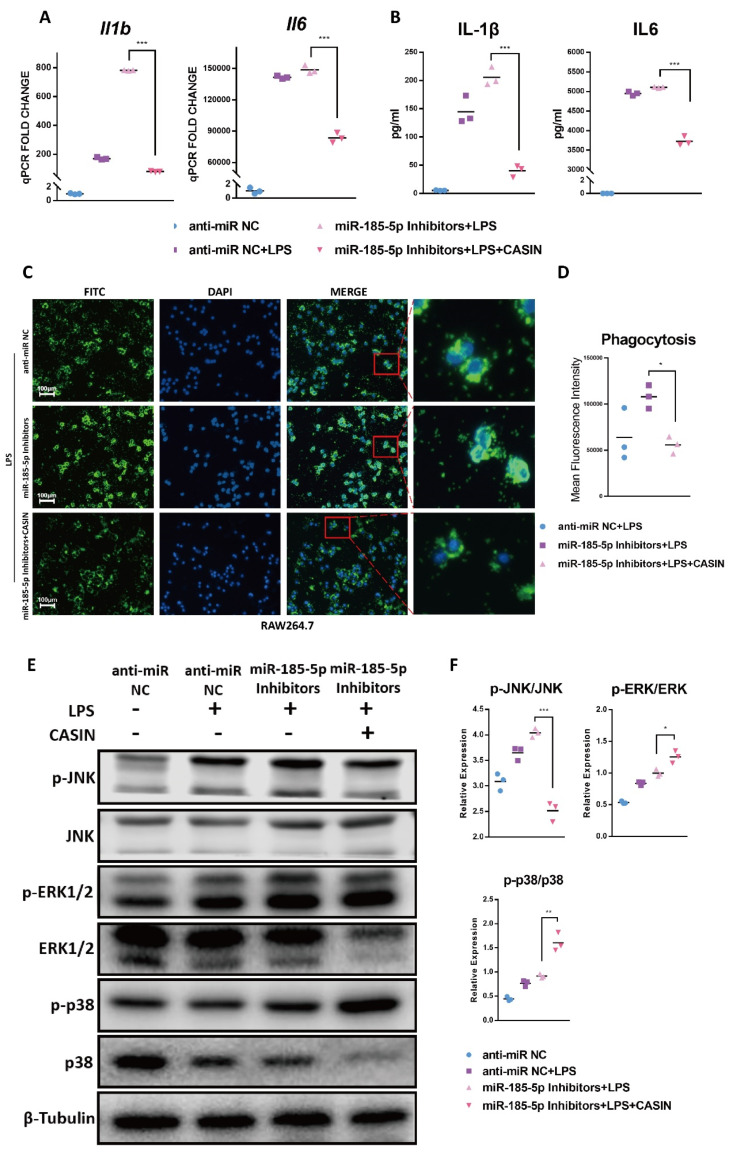
CASIN modulates miR-185-5p-mediated pro-inflammatory phenotype partially through targeting MAPK JNK pathways. (**A**) Gene expressions (RT-qPCR) and (**B**) cytokine secretions (ELISA) in LPS-induced RAW264.7 macrophages with 150 nM miR-185-5p inhibitors or anti-miR NC transfection, 1 μM CASIN and DMSO treatment. (**C**) Representative images showing phagocytosis of fluorescein-labeled *E. coli* bioparticles (FITC-Green, DAPI-Blue) in LPS-induced RAW264.7 macrophages with 150 nM miR-185-5p inhibitors or anti-miR NC transfection, 1 μM CASIN and DMSO treatment. The scale bars correspond to 100 μm. (**D**) Quantification of mean fluorescence intensity of phagocytosis. (**E**) Effects of CASIN treatment and miR-185-5p knockdown on the activation of the total and phosphorylation of critical proteins in MAPK pathways after treatment with or without 100 ng/mL LPS for 24 h and 1 μM CASIN or DMSO, as determined by Western blot. (**F**) Quantification of phosphorylation and total proteins in Western blots. (**G**) cytokine secretions (ELISA) in LPS-induced RAW264.7 macrophages with 150 nM miR-185-5p inhibitors or anti-miR NC transfection, 1 μM, 5 μM SP600125 and DMSO treatment. Mean ± SD; *n* = 3; * *p* < 0.05; ** *p* < 0.01; *** *p* < 0.001.

**Figure 7 genes-13-00468-f007:**
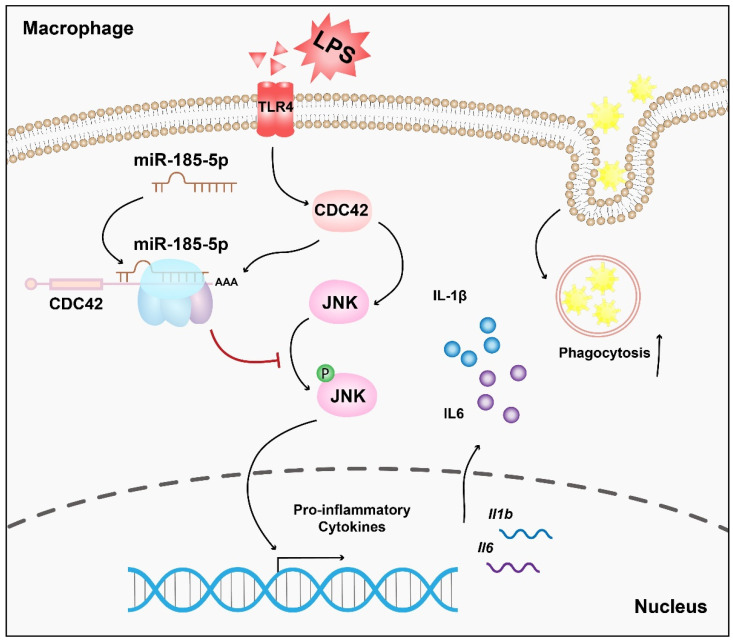
Schematic illustration of the mechanism in which miR-185-5p regulates inflammation and phagocytosis in LPS-induced RAW264.7 macrophages.

**Table 1 genes-13-00468-t001:** Detailed information of the GSE87718 and GSE143845 datasets.

Accession	Platform	Sample	Control Group	Treated Group
GSE87718	GPL18082	6	Dicer Wild Type BMDMs (*n* = 3)	Dicer Knock Out BMDMs (*n* = 3)
GSE143845	GPL21265	6	RAW264.7 untreated (*n* = 3)	RAW264.7 treated with LPS (100ng/mL) and INF-γ (20ng/mL) for 24 h (*n* = 3)

**Table 2 genes-13-00468-t002:** List of sequences used in the study.

RT-qPCR		
*Il6*	Forward Primer	acaaagccagagtccttcagag
*Il6*	Reverse Primer	accacagtgaggaatgtccac
*Il1b*	Forward Primer	tgccaccttttgacagtgatg
*Il1b*	Reverse Primer	tgatactgcctgcctgaagc
*Tnf*	Forward Primer	tgttgcctcctcttttgctt
*Tnf*	Reverse Primer	tggtcaccaaatcagcgtta
*Cdc42*	Forward Primer	cggagaagctgaggacaagatctaa
*Cdc42*	Reverse Primer	aggagacatgttttaccaacagc
*Actb*	Forward Primer	accttctacaatgagctgcg
*Actb*	Reverse Primer	ctggatggctacgtacatgg
*U6*	Forward Primer	ctcgcttcggcagcaca
*U6*	Reverse Primer	aacgcttcacgaatttgcgt
mmu-miR-185-5p	gtcgtatccagtgcagggtccgaggtattcgcactggatacgactcagga	
**miRNA transfection**		
miR NC	Sense (5′-3′)	UUUGUACUACACAAAAGUACUG
miR NC	Anti-sense (5′-3′)	CAGUACUUUUGUGUAGUACAAA
miR-185-5p mimics	Sense (5′-3′)	UGGAGAGAAAGGCAGUUCCUGA
miR-185-5p mimics	Anti-sense (5′-3′)	UCAGGAACUGCCUUUCUCUCCA
Anti-miR NC	CAGUACUUUUGUGUAGUACAAA	
miR-185-5p inhibitors	UCAGGAACUGCCUUUCUCUCCA	

## Data Availability

All datasets generated and analyzed in this study are available from the corresponding author on reasonable request. GEO datasets analyzed in this study are available at https://www.ncbi.nlm.nih.gov/geo/.

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
