# Peer review of "miR-185-5p Regulates Inflammation and Phagocytosis through CDC42/JNK Pathway in Macrophages"

_genes, 2022, doi:10.3390/genes13030468_

Round 1

Reviewer 1 Report

The research article entitled, “miR-185-5p regulates inflammation and phagocytosis through CDC42 /JNK pathway in macrophages" by Ma et al., aims to understand the guiding role of miR-185-5p in macrophage inflammatory activation. They identified that miR-185-5p was one of the top genes and effectively down-regulated in two macrophages miRNA arrays from GEO database and miR-185-5p sponged and negatively regulated the protein expression of CDC42. Authors further demonstrated that miR-185-5p exhibited anti-inflammatory functions in LPS-induced RAW264.7 macrophages at least partially through CDC42/JNK pathways.

Altogether this is an important and timely research article, this reviewer has certain suggestions that would help produce a more comprehensive overview of the topic:

Comments:

1, Figure 1 quality may be improved (high resolution).

2, The English of manuscript can be polished (minor).

3, Authors can add one paragraph for abbreviations.

4, Briefly discuss the translational efficacy of their study and future direction to this area of macrophage inflammation.

5, Authors can include the limitations to their study.

6, Delete/remove lines 410-411 (This section is not mandatory but can be added to the manuscript if the discussion is unusually long or complex.).

7, Did authors find any other targets of miR-185-5p? discuss briefly.

Author Response

Dear reviewer,

Thank you very much for the suggestions.

We uploaded Figure1 with higher quality(1200dpi). Abbreviations in this manuscript were listed in Supplementary Table1. English Errors and grama mistakes were removed and edited.

We discussed our study limitations in the last paragraph of the Discussion. The significance of this study and research expectations were added in the Conclusion.

We did find another target of miR-185-5p and briefly discussed in lines 396-399. We are planning experiments to verify this interaction and functions in further work.

With regards,

Fengling Chen

Feb 23rd 2022

Reviewer 2 Report

Ma et al tried to investigate miR-185-5p as one of the top genes which is effectively down-regulated in two macrophages miRNA arrays from GEO database. The author demonstrated that miR-185-5p exhibited anti-inflammatory functions in LPS-induced RAW264.7 macrophages at least partially through CDC42/JNK pathways. Further author revealed that these findings could yield insights to understand miRNA regulation network in macrophage inflammation. The present study added some interesting information to the research area in this field; however, I have few concerns regarding this study as below-

  1. I wonder whether author performed any experiment regarding macrophage foam cell formation which is an important part of atherosclerosis and whether this miR-185-5p have any effect on genes involved in foam cell formation such as PPAR-Y, CD38?
  2. Did author take polarization of macrophages M1 M2 into consideration while designing these experiments.
  3. Author should add relevant references in material and methods such as quantification of RT-PCR and wherever necessary.
  4. Please proof read the manuscript carefully e.g L410 should be removed.

Author Response

Dear Reviewer,

Thank you so much for the helpful suggestions.

Since miR-185-5p was screened out as one of the top genes under the background of macrophage inflammation and atherosclerosis miRNA expression profiles, we mainly focused on the inflammatory effect of miR-185-5p in macrophage. Foam cell formation is essential for the development of atherosclerosis, especially at the early stage. miR-185-5p is related to cholesterol metabolism in mouse liver (Jiang et al. doi:10.1016/j. Atherosclerosis.2015.10.026.). Your suggestion is very helpful for us in future research and experiment design.

Classically M1 activated phenotype plays important role in atherosclerotic inflammation. We have tried to stimulate RAW264.7 cells with LPS&INF-γ, which is a classical M1 polarization cell line model. In response to LPS&INF-γ for 24 hours, the inhibitory effect of combined treatment on miR-185-5p was similar to LPS, therefore we gave up the combined stimulation. In addition, the interaction between miR-185-5p and M2 polarization has been discussed in kidney stone and kidney injuries. They indicate macrophage M2 polarization is associated with the miR-185-5p/CSF-1 signals (Zhu et al. doi: 10.1038/s41419-019-1358-y. Cell Death Dis. 2019 Mar 20).

Relevant references were added in material and methods. Errors and mistakes were removed and edited.

With regards,

Fengling Chen

Feb 23rd 2022